



# A Novel Method to Quantify the Uncertainty Contribution of Aerosol-Radiative Interaction Factors

Bishuo He[1], Chunsheng Zhao[1]

[1]Department of atmospheric and oceanic sciences, School of physics, Peking University, Beijing, 100080, China

*Correspondence to*: Chunsheng Zhao (zcs@pku.edu.com)

**Abstract.** The IPCC's assessment report shows that the radiative forcing of aerosol-radiation interactions still involves significant uncertainty. The commonly used method for factor uncertainty estimation is the One-at-A-Time (OAT) method which evaluates factor sensitivity by controlling the change in a single variable while keeping others constant. The outcomes from the OAT method require high data quality to ensure accuracy, and the results are only valid near the selected constant.

This study proposes a new method called Constrained Parameter (CP) to quantify the uncertainty contribution of factors in a multi-factor system. This method constrains the uncertainty of a single factor and evaluates its sensitivity by analyzing how this change affects output uncertainty. The most significant advantage of the CP method is that it can be applied to any data distribution, and its results can reflect the overall data characteristics. By comparing the results calculated by the CP method and the OAT method, the proportion of factor interactions in the factor uncertainty contributions can be obtained. As an

application of the CP method, it is used to perform a detailed analysis of aerosol-radiation interaction factors' uncertainty contributions. The top 3 most sensitive factors are the complex refractive index of aerosol shell materials, light-absorbing carbon parameters, and Mie theory parameters. Due to their high sensitivity and low observational precision, these factors represent significant sources of uncertainty in aerosol-radiation interactions. These factors need to be prioritized for operational observation programs and model parameter inputs.

## 1 Introduction

Aerosol-Radiation Interaction (ARI) refers to the direct scattering and absorption of solar radiation by aerosols, and it's a key component of aerosol radiative forcing that can have a significant influence on the climate system (IPCC, 2021). In recent years, numerous studies have focused on $RF_{ari}$ and its associated impacts. According to the Intergovernmental Panel on Climate Change (IPCC) Sixth Assessment Report (Forster et al., 2021), the estimated $RF_{ari}$ is $-0.22[-0.47\sim0.04]$W/

$m^2$, a value comparable to the radiative forcing of $N_2O$ at $0.21[0.18\sim0.24]$W/m$^2$, and second only to that of $CO_2$ at $2.16[1.90\sim2.41]$W/m$^2$, $CH_4$ at $0.54[0.43\sim0.65]$W/m$^2$, and $O_3$ at $0.47[0.24\sim0.71]$W/m$^2$. Despite significant advances, the estimated $RF_{ari}$ has undergone substantial revisions across successive IPCC reports, and its uncertainty has not notably decreased (Houghton, 1996; Houghton, 2001; Solomon, 2007; Stocker, 2014; Forster et al., 2021). Furthermore, significant discrepancies exist between $RF_{ari}$ estimates derived from surface temperature changes and those from model simulations





(Anderson et al., 2003; Hansen et al., 2023), suggesting that the cooling effect of aerosols may be underestimated, potentially due to the omission of key factors in current models. To address these issues, it is crucial to conduct a thorough and precise uncertainty analysis of ARI factors and improve routine aerosol observation projects and model settings to better capture the contributions of critical factors.

At present, research on assessing the uncertainty contributions of ARI factors has primarily focused on the sensitivity of

these factors (McComiskey et al., 2008; Loeb and Su, 2010; Lee et al., 2011; Srivastava et al., 2011; Lee et al., 2016). The commonly used sensitivity analysis method is the One-at-a-time (OAT) method, which is based on the principle of the control variable method. In this approach, a single factor $dx$ is varied while all other factors are held constant, and the corresponding change in the output $dy$ is observed. The sensitivity of a factor is then defined as

$$S_{OAT} = \frac{dy}{dx}, \tag{1}$$

The main advantage of the OAT evaluation method is its computational efficiency, as it requires only two experiments to determine the sensitivity of a factor, making it particularly suitable for large-scale ensemble models where computational costs are high. To ensure the accuracy of the OAT method, two conditions must be met: first, the covariance among the observed data of each factor must be zero; second, the sensitivity results obtained are only valid near the selected constant value. For ARI system, these conditions may not be strictly fulfilled, a discussion that will be elaborated upon in section 2.

In addition to the possible errors caused by the evaluation method, the factors discussed in the current assessment work exhibit certain limitations. In order to facilitate the discussion in the model, evaluation work usually only focuses on the most common observation items: Aerosol Optical Parameters (AOPs), including Aerosol Optical Depth (AOD), Single Scattering Albedo (SSA), and Asymmetry Factor (g). Many works have shown that AOPs are the most direct influencing factor of $RF_{ari}$ (Andrews et al., 2006; McComiskey et al., 2008; Loeb and Su, 2010; Chung, 2012; Zhao et al., 2018). AOPs

are affected by various aerosol parameters and environmental parameters, which in turn affect $RF_{ari}$, but there is little discussion on the uncertainty contribution of such factors. The lack of a complete evaluation of the factors may lead to the importance of some factors being underestimated. Moreover, in assessment reports from the IPCC and similar studies, multiple global models are utilized to independently simulate the radiative forcing effects. The parameterization processes in these large models simplify the underlying physical mechanisms, and the differing settings among models hinder a

comprehensive understanding of ARI. As a result, this variability can lead to inaccurate conclusions regarding the effects of ARI.

This study introduces a new method for analyzing the uncertainty contribution of factors in multi-factor system. As an application, $RF_{ari}$ is calculated using a radiative transfer mechanism model and the uncertainty contribution of the factors is analyzed. The factors identified as high uncertainty contribution through this analysis should be prioritized in future aerosol

observational projects and model settings to prevent overlooking significant influences. Section 2 provides a detailed introduction to the new method, while Section 3 presents the analysis results for ARI factor uncertainty contributions and compares them with those obtained using the OAT method.



## 2 Analysis Method of Factor Uncertainty Contributions

This section will provide a comprehensive analysis of the most commonly used method for assessing factor uncertainty contributions through control variables. Additionally, it will introduce a new method for analyzing factor uncertainty contributions, aiming to address potential issues associated with the traditional control variable method.

### 2.1 OAT Method

For a multi-factor system, it can be expressed as

$$y = f(X), \tag{2}$$

Among them, $y$ is the output variable, $X$ is the input variable, $X = [x_1, x_2, \dots, x_n]^T$. Since there may be interactions between factors, the input variables satisfy

$$x_i = g_i(X), \tag{3}$$

$$\frac{\partial X}{\partial x_i} = [\frac{\partial g_1}{\partial x_i}, \frac{\partial g_2}{\partial x_i}, \dots, \frac{\partial g_n}{\partial x_i}]^T, \tag{4}$$

If the variables are independent of each other, then

$$g_i(X) = x_i, \tag{5}$$

Taylor expansion of this formula gives

$$f(X) = f(A) + [\nabla f(A)]^T(X - A) + \frac{1}{2!}[X - A]^T H(A)[X - A] + \cdots, \tag{6}$$

Where $H(A)$ is the Hessian matrix of $f(X)$ at $A$. When $X \to A$ is satisfied, the higher-order terms above the second order tend to 0, and at this time

$$f(X)_{X \to A} = f(A) + [\nabla f(A)]^T(X - A) + O^2, \tag{7}$$

If the $X \to A$ condition cannot be met, the influence of the higher-order terms in the equation cannot be ignored. Taking the partial derivative of $f(X)$ with respect to the input variable $x_i$, we have

$$\frac{\partial f(X)}{\partial x_i}_{X \to A} = \frac{\partial [\nabla f(A)]^T(X-A)}{\partial x_i} = \sum_{j=1}^{n} \frac{\partial f(A)}{\partial x_j} \frac{\partial g_j(X)}{\partial x_i}, \tag{8}$$

When the variables are independent of each other,

$$\frac{\partial g_j(X)}{\partial x_i} = 1, j = i, \tag{9}$$

$$\frac{\partial g_j(X)}{\partial x_i} = 0, j \neq i, \tag{10}$$

The sensitivity analysis results of the OAT method can be obtained

$$S_{OAT} = \frac{\partial f(X)}{\partial x_i}_{X \to A} = \frac{\partial f(A)}{\partial x_i}, \tag{11}$$

There is a linear relationship between the output variable and all variables

$$y|X \to A = KX + C, \tag{12}$$



Where $K = [\nabla f(A)]^T$, $C = f(A) - A[\nabla f(A)]^T$, and the uncertainty of each input variable and output variable has a transfer relationship

$$D_{\sigma_y^2} = K^T D_{\sigma_{x_i}^2} K, \tag{13}$$

Here, $\sigma_y$ and $\sigma_{x_i}$ are the standard deviations of the output variable and each input variable, respectively, and $D$ represents a

diagonal matrix. Under these conditions, the uncertainty of the output can be decomposed into factor sensitivity and factor uncertainty, expressed as follows:

$$U_y^2 = \sum S_i^2 \times U_{x_i}^2, \tag{14}$$

When the variables are not independent, we have

$$S_{OAT} - \frac{\partial f(A)}{\partial x_i} = \sum_{j=1, j \neq i}^n \frac{\partial f(A)}{\partial x_j} \frac{\partial g_j(X)}{\partial x_i}, \tag{15}$$

It can be concluded that the sensitivity analysis results obtained by the OAT method are valid only when:

(1) $X \to A$;

(2) The variables are independent of each other.

When these two conditions are met, the analysis results of the OAT method can strictly reflect the sensitivity of the input variable $x_i$ to the output $y$.

## 105 2.2 Applicability of OAT Method in ARI system

For the ARI system, meeting these two conditions is challenging for the following reasons:

(1) In aerosol observations, inaccuracies in certain instruments or uncertainties arising from the inversion process of the joint observation system can lead to significant uncertainties in the generated observation data. As a result, the statistical average of the measurement results may not accurately reflect the true properties of aerosols. This discrepancy means

that there can be a considerable deviation between the calculated $A$ value $A_{cal}$ and the actual $A$ value $A_{real}$, resulting in obtained results that are not entirely accurate.

(2) Due to the influence of various factors such as aerosol sources, aging processes, and changes in the atmospheric environment, the observed values of the physical and chemical properties of aerosols may vary significantly over time. The sensitivity analysis conducted using the OAT method on a set of observational data only reflects conditions near the

selected $A$ value, meaning that the sensitivity results obtained cannot adequately represent the overall situation.

(3) Considering the temporal variations (such as diurnal, seasonal, and interannual changes) and spatial variations (including differences between coastal and inland areas, urban and rural settings, and between the boundary layer and free atmosphere), the observed data may exhibit a linear trend rather than conforming to a strict normal distribution. When performing sensitivity analysis, it is essential to first account for these variation trends in the data; otherwise, the

results may be distorted. To address this issue, a thorough analysis of the actual physical environment is required, which may necessitate additional time and computational resources.



(4) For ARI, the interactions between factors are significant, making it impossible to ensure that all discussed factors are strictly independent of one another; thus, the covariance between observed data may not equal zero. When applying the OAT method to analyze the sensitivity of a specific factor, it is essential to hold the values of other factors constant. However, this approach does not accurately reflect the actual physical conditions, leading to errors in the final results related to factor interaction terms.

(5) When evaluating $RF_{ari}$ in a large ensemble model, certain factors are often parameterized due to constraints related to computational costs. This parameterization can lead to discrepancies between the physical processes represented in the model and the actual conditions, meaning that the results obtained from sensitivity analyses may not accurately reflect reality. Additionally, some parameterized settings may introduce correlations between factors, further compromising their independence and affecting the overall reliability of the analysis.

The outcome of sensitivity analysis is to ascertain the uncertainty contribution of various factors. It is essential to perform separate statistical analyses on the sensitivity of these factors and on observational uncertainty. To address the challenge of high uncertainty in the evaluation results of $RF_{ari}$, a more practical discussion should focus on how constraints imposed by observations, or improvements in observational accuracy, can enhance the reliability of the results. While sensitivity analysis does not directly answer this question, it offers valuable insights into the significance of each factor from a different perspective, guiding future efforts to reduce uncertainty.

In summary, when employing the OAT method to analyze factor sensitivity within the ARI system, there are strict requirements and numerous limitations regarding data quality. Forced application of this method may lead to discrepancies between the results and actual conditions. Besides the OAT method, various sensitivity analysis techniques have been widely utilized across many fields (Hamby, 1995; Christopher and Patil, 2002; Saltelli et al., 2005; Marino et al., 2008). Each of these methods also has specific requirements for data quality. Therefore, to enhance the reliability of the evaluation results for $RF_{ari}$, it is crucial to adopt a more suitable analysis method to assess the uncertainty contributions and significance of the influencing factors.

## 2.3 CP Method

Building on the history match method (Edwards et al., 2011; Williamson et al., 2013; Lee et al., 2016), this work introduces the Constrained Parameter (CP) method to quantitatively rank the uncertainty contributions of various factors. The central concept of the CP method is to define the importance of a variable's uncertainty by constraining the value range of the input variable and observing how this constraint affects the standard deviation of the output variable. The specific analytical approach includes the following steps:

(1) A Monte Carlo (MC) simulation is conducted on the system, where the range of each input parameter is determined based on the distribution of actual measurement results. The initial MC simulation allows for the exploration of all possible states of the system given the input variable distribution, enabling the calculation of the standard deviation $\sigma_y$



of the output variable. To enhance the efficiency of the MC simulation, Latin Hypercube Sampling (LHS) is employed

for data sampling.

(2) For a specific input variable, the distribution range is constrained, reducing its standard deviation from $\sigma_x$ to $\sigma_x{}'$, while keeping the standard deviation of the other input variables unchanged. However, when the factors are not independent, constraining the distribution of one input variable may also alter the distributions of the other input variables.

(3) Conduct another Monte Carlo (MC) simulation using the updated range of input variables to obtain the new standard

deviation $\sigma_y{}'$ of the output variable.

(4) The sensitivity of the input variable to the output variable is defined as:

$$S_{CP} = \sqrt{\frac{\sigma_y{}^2 - \sigma_y{}'^2}{\sigma_x{}^2 - \sigma_x{}'^2}} = \sqrt{\frac{d\sigma_y{}^2}{d\sigma_x{}^2}}, \tag{16}$$

Since only a specific input variable is constrained while the standard deviations of the distributions of the other input variables remain unchanged, the change in the distribution of the output variable is solely influenced by the alteration in the

uncertainty of the constrained input variable. Therefore, the definition of $S_{CP}$ is specifically related to the uncertainty of the data, and it can also be referred to as the sensitivity of uncertainty. This method is not limited to specific system equations and can be applied to all multi-factor systems.

When the factors are strictly independent of each other, the covariance between the observed data is equal to zero, which can be expressed mathematically as:

$$cov(x_i, x_j) = E(x_i x_j) - E(x_i)E(x_j) = 0, \tag{17}$$

Satisfying the error transfer formula, we obtain:

$$\sigma_y{}^2 = \frac{\partial f(X)^2}{\partial x_1} \cdot \sigma_{x_1}{}^2 + \cdots + \frac{\partial f(X)^2}{\partial x_i} \cdot \sigma_{x_i}{}^2 + \cdots + \frac{\partial f(X)^2}{\partial x_n} \cdot \sigma_{x_n}{}^2, \tag{18}$$

For the CP method, we can get

$$\sigma_y{}'^2 = \frac{\partial f(X)^2}{\partial x_1} \cdot \sigma_{x_1}{}^2 + \cdots + \frac{\partial f(X)^2}{\partial x_i} \cdot \sigma_{x_i}{}'^2 + \cdots + \frac{\partial f(X)^2}{\partial x_n} \cdot \sigma_{x_n}{}^2, \tag{19}$$

$$d\sigma_y{}^2 = \frac{\partial f(X)^2}{\partial x_i} \cdot d\sigma_{x_i}{}^2, \tag{20}$$

$$S_{CP} = \sqrt{\frac{d\sigma_y{}^2}{d\sigma_x{}^2}} = \frac{\partial f(X)}{\partial x_i} = S_{OAT}, \tag{21}$$

Thus, when the factors are independent of one another, the sensitivity analysis results obtained using the CP method will align with those obtained through the OAT method. Consequently, the difference between the results of the CP method and the OAT method highlights the effects of factor interactions, providing insights into how these interactions may affect factor

uncertainty contributions.

The advantages of the CP method are as follows:

(1) To ensure accuracy, the OAT method must strictly satisfy the condition of $X \to A$, meaning that the sensitivity analysis results are only valid near the value of $A$. In contrast, the CP method examines the relationship between factor



uncertainty and output uncertainty. The sensitivity results derived from this method reflect the overall data distribution rather than focusing on a specific fixed value. Consequently, the analysis results obtained from the CP method are more representative and applicable.

(2) When the distribution of one factor is constrained, it influences the distributions of other interacting factors as well. Consequently, any changes to the constraints of the input variable will also lead to alterations in the distributions of these other variables. Different constraints include two categories: one is the difference in data distribution types, such as uniform distribution and normal distribution; the other is the difference in data standard deviation, which is reflected in different degrees of improvement in observation accuracy. As a result of this influence, the output distribution will be affected by the level of constraint applied to the factor, leading to different sensitivity outcomes. Therefore, the results of the CP method can reflect the impact of different data constraints on the uncertainty of the output, which cannot be reflected in the OAT method.

(3) The analytical method is used to obtain the quantitative ranking results of factor uncertainty contributions, aiming to solve the problem of large output uncertainty by focusing on high uncertainty contribution factors. The sensitivity analysis result $S_{CP}$ illustrates the relationship between factor uncertainties and output uncertainties. The resulting ranking provides a direct response to this issue. The high uncertainty contribution factors identified by the CP method are essential elements in observation and model design that require our attention and enhancement. In contrast, the results from the OAT method represent statistical sensitivity concepts, serving as indirect indicators of factor uncertainty contributions, and lack robust physical grounding.

Therefore, the CP method is employed to quantify the contribution of factor uncertainty, and the results obtained offer a more accurate representation of each factor's importance in practical physical terms. This approach enhances our understanding of how various factors influence the overall uncertainty, enabling more informed decision-making in observation projects and model development.

## 3 Calculation of Uncertainty Contribution of $RF_{ari}$ Factors

In this section, the uncertainty analysis method proposed in this paper is applied to the ARI system to verify its feasibility. A detailed discussion is provided regarding the importance of each factor within the ARI system, highlighting how this method enhances our understanding of their contributions to overall uncertainty.

### 3.1 Analysis of ARI Factors

Aerosol particles in the environment exhibit complex characteristics influenced by multiple factors. To effectively simulate the ARI system and minimize excessive parameterization, this study employs the radiative transfer mechanism model to calculate $RF_{ari}$. The model used is SBDART (Santa Barbara DISORT Atmospheric Radiative Transfer), developed by Ricchiazzi et al. (1998), which is capable of simulating and calculating radiation processes involving aerosols, the





atmosphere, surfaces, clouds, and solar spectra. Additionally, the Mie theory (Mie, 1908) is utilized to characterize the radiative properties of individual aerosol particles, allowing for the calculation of aerosol optical parameters (AOPs) such as aerosol optical depth (AOD), single scattering albedo (SSA), and asymmetry factor (g). These parameters serve as critical input for the SBDART model. The results of the factor analysis are presented in Figure 1.

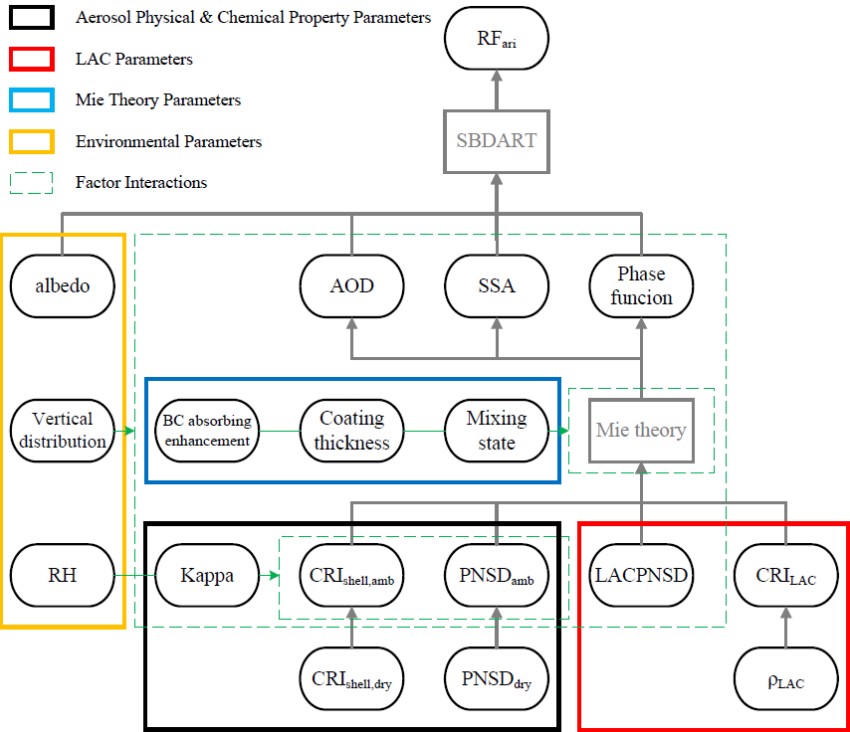

**Figure 1: Analysis of ARI factors.**

In this analysis, we adopt a core-shell model assumption for aerosol particles, assuming that the core-shell composition is uniform. Specifically, the shell material is primarily composed of scattering materials, while the core consists of light-absorbing carbon (LAC). The radiative characteristics of individual aerosol particles are determined by the complex refractive index of the core-shell material, $CRI_{shell,dry}$, $CRI_{LAC}$, and the size of the core-shell particles. LAC is modeled as a

combination of dense elemental carbon (EC) and hollow air (Zhao et al., 2020). By measuring the densities of EC and LAC, denoted as $\rho_{EC}$ and $\rho_{LAC}$, we can calculate the complex refractive index of LAC through a weighted approach.

For a group of particles, it is essential to consider the size spectrum distribution of the aerosol particles and the LAC size spectrum distribution, $PNSD_{dry}$ and $LACPNSD$, respectively, to perform weighted calculations of the radiative characteristics of the aerosol ensemble. Additionally, aerosol particles absorb moisture from the environment, influenced by

the hygroscopic parameter (Kappa) and the ambient relative humidity (RH). This moisture absorption leads to changes in the complex refractive index of the shell material, $CRI_{shell,amb}$, and alters the ambient particle size spectrum distribution, $PNSD_{amb}$.



The strict assumptions of Mie theory often diverge from real-world conditions, resulting in discrepancies between theoretical calculations and observed phenomena. Numerous studies have explored these discrepancies (Volten et al., 2001; Cappa et al., 2012; Fierce et al., 2016; Zhang et al., 2017; Freedman, 2020). To better understand the impact of these Mie theory assumptions on $RF_{ari}$, we will analyze the uncertainty contributions of several key factors, including the mixing state of aerosols ($MS$), LAC absorbing enhancement ($LACAE$), and coating thickness ($CT$).

In the real atmospheric environment, aerosols and environmental parameters exhibit a vertical profile distribution, leading to significant variations in aerosol radiative capabilities at different altitudes. To investigate the impact of different vertical distribution types (VD) on $RF_{ari}$, we employ the vertical distribution parameterization scheme developed by Liu et al. (2009), which is grounded in aircraft observations. Using this scheme, we calculate the vertical distribution of $PNSD_{amb}$ and $LACPNSD$. Additionally, the vertical distribution of environmental parameters is established based on reanalysis data.

According to the categories, all factors are divided into four categories:

(1) Aerosol physical and chemical property parameters, including $CRI_{shell,dry}$, $PNSD_{dry}$, and $Kappa$;

(2) LAC parameters, including the real part $n_{LAC}$ and imaginary part $k_{LAC}$ of $CRI_{LAC}$, $\rho_{LAC}$, and $LACPNSD$;

(3) Mie theory parameters, including $MS$, $CT$, and $LACAE$;

(4) Environmental parameters, including $VD$, $RH$, and $albedo$.

The uncertainty contributions of the four types of factors are discussed separately to determine the type of factors with the strongest impact.

All the factors depicted in the Fig.1 influence $RF_{ari}$. We evaluate the uncertainty contributions of each factor using both the OAT method and the CP method, comparing the differences between the two approaches. Additionally, since AOD, SSA, and g are direct input parameters for SBDART and have the most immediate effects on $RF_{ari}$, we also discuss the sensitivity of each factor in relation to these AOPs.

## 3.2 Data Sources and Mode Environment Settings

In the simulation experiment, aerosol and environmental parameters are derived from a combination of field observations, previous research summaries, and instrument observation network data. Typical aerosol and environmental data representative of North China are utilized for the simulation. The calculation of $RF_{ari}$ incorporates integrated results across the full solar spectrum (0.25–4.00 μm), focusing specifically on the instantaneous results at noon on the summer solstice. Aerosol data are sourced from the Peking University observation station (116°W, 40°N) and the Taizhou observation station (120°W, 33°N). Environmental data are drawn from the fifth-generation ECMWF reanalysis data (ERA5) provided by the European Centre for Medium-Range Weather Forecasts (ECMWF), as well as Moderate Resolution Imaging Spectroradiometer (MODIS) observation data collected from the Terra and Aqua satellites. The details are summarized in Table 1.

| Properties | Values | Data Source |
|---|---|---|




| | | |
|---|---|---|
| $CRI_{shell}$ | 1.58+1e-3i | Peking University Observation Site |
| $CRI_{EC}$ | 2.26+1.26i | Taylor et al., 2015 |
| $CRI_{air}$ | 1+1e-3i | Zhao et al., 2020 |
| $\rho_{EC}$ | 1.8 | Bond and Bergstrom, 2006 |
| $\rho_{LAC}$ | 0.95 | Zhao et al., 2019 |
| $Kappa$ | 0.215 | Peking University Observation Site |
| $MS$ | 0.7 | Gong et al., 2016 |
| $PNSD_{dry}$ | / | Taizhou Observation Site |
| $LACPNSD$ | / | Taizhou Observation Site |
| $VD$ | / | Liu et al., 2009 |
| $RH$ | / | ERA5 Data |
| $albedo$ | / | MODIS Data |
| Location Settings | 116° W, 40° N | / |
| Date Settings | April 1 | / |
| Time Settings | 12:00 local time | / |

**Table 1: Observation data sources and SBDART environment parameter settings.**

265 **3.3 Quantitative Ranking Results of ARI Factor Uncertainty Contributions**

The sensitivity analysis of the factors affecting $RF_{ari}$ and AOPs was conducted using both the OAT method and the CP method. The results of this analysis are illustrated in Figure 2 and Figure 3.

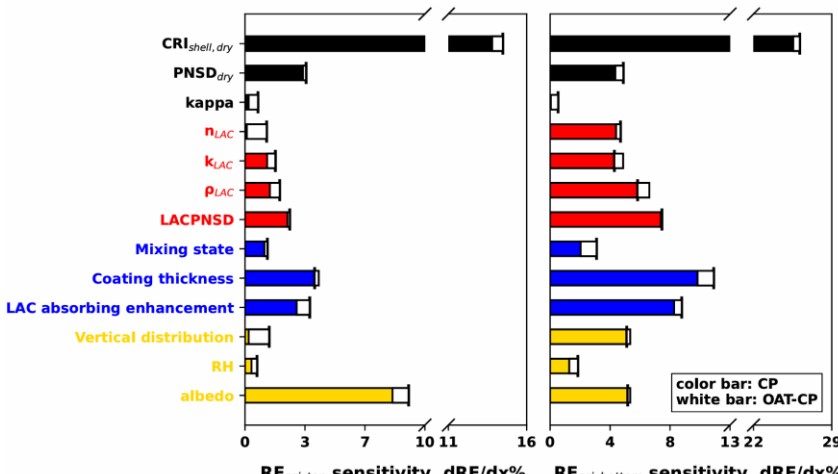





**Figure 2: The factor uncertainty contribution analysis results of $RF_{ari}$. The results from the CP method are represented by color bars, while the differences between the CP method and the OAT method are indicated by white bars.**

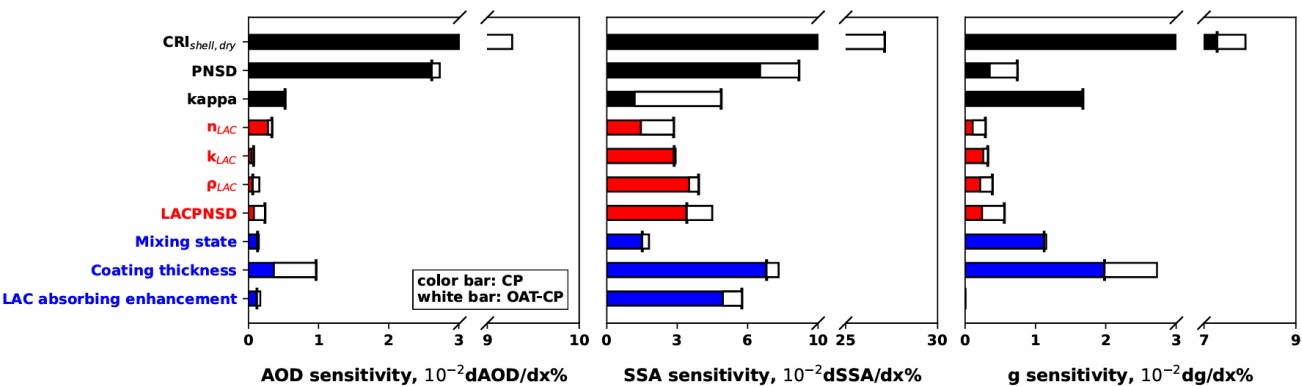

**Figure 3: The factor uncertainty contribution analysis results of AOPs. The results from the CP method are represented by color bars, while the differences between the CP method and the OAT method are indicated by white bars.**

### 3.3.1 Comparison of Results between CP Method and OAT Method

The sensitivity analysis results indicate that the OAT method overlooks the impact of factor interactions, while the CP method accounts for these interactions. Consequently, the differences between the results from the two methods provide a measure of the influence of factor interactions on the sensitivity outcomes. In the analysis of $RF_{ari,top}$ and $RF_{ari,bottom}$, the proportion of interaction effects ranges from 1.1% to 91%, with an average of 25%. After weighting according to sensitivity, the average difference is calculated to be 10%. For AOPs, the proportion of interactions varies from 0.33% to 386%, with an average of 56% and a weighted average of 25%. This variability suggests that different factors experience varying degrees of influence from interactions, with some factors significantly affected. Therefore, using the OAT method for sensitivity analysis may lead to an average relative error of 10% for $RF_{ari}$ and 25% for AOPs, due to the neglect of factor interactions.

The CP method enables sensitivity analysis for data with varying degrees of factor constraints and non-normal distributions. The OAT method's sensitivity results are only valid near the chosen $A$ value, meaning that changes in the A value may significantly impact the results. Additionally, the choice of $dx$ in the $\frac{dy}{dx}$ calculation can also influence outcomes. To explore these aspects, we computed results for the following seven sensitivity analysis cases:

(1) Using the CP method, when the data is normally distributed, constrain the data distribution of a certain factor so that the standard deviation is reduced from $\sigma_{x_i}$ to $\sigma_{x_i}'$, and perform sensitivity analysis;

(2) Using the CP method, when the data is normally distributed, constrain the data distribution of a certain factor so that the standard deviation is reduced from $\sigma_{x_i}$ to $2\sigma_{x_i}'$, and perform sensitivity analysis;

(3) Using the CP method, when the data is uniformly distributed, constrain the data distribution of a certain factor so that the standard deviation is reduced from $\sigma_{x_i}$ to $\sigma_{x_i}'$, and perform sensitivity analysis;

(4) Using the OAT method perform sensitivity analysis;





(5) Using the OAT method, set the $A$ value as 0.9 times that of case (4), perform sensitivity analysis;

(6) Using the OAT method, set the $A$ value as 1.1 times that of case (4), perform sensitivity analysis;

(7) Using the OAT method, set the $dx$ value as 0.5 times that of case (4), perform sensitivity analysis;

(8) Using the OAT method, set the $dx$ value as 2 times that of case (4), perform sensitivity analysis.

A sensitivity analysis of $RF_{ari,top}$ was performed on all the above situations, and the results are shown in Table 2.

| | $CP_{normal-normal}$ | $CP_{normal-normal}'$ | $CP_{uniform-normal}$ | $OAT$ | $OAT_{0.9A}$ | $OAT_{1.1A}$ | $OAT_{0.5dx}$ | $OAT_{2dx}$ |
|---|---|---|---|---|---|---|---|---|
| $CRI_{shell,dry}$ | 14.8 | 15.7 | 14.6 | 14.2 | 8.90 | 19.3 | 14.9 | 14.08 |
| $PNSD_{dry}$ | 3.31 | 4.49 | 4.41 | 3.15 | 3.20 | 3.05 | 3.20 | 3.15 |
| $\kappa$ | 0.71 | 0.59 | 0.37 | 0.20 | 0.20 | 0.15 | 0.20 | 0.18 |
| $n_{BC}$ | 1.17 | 1.19 | 1.20 | 0.10 | 0.30 | 0.10 | 0.40 | 0.20 |
| $k_{BC}$ | 1.65 | 1.99 | 0.90 | 1.20 | 1.40 | 1.45 | 1.60 | 1.43 |
| $\rho_{BC}$ | 1.88 | 1.60 | 1.46 | 1.35 | 1.70 | 1.50 | 0.80 | 1.60 |
| $LACPNSD$ | 2.43 | 2.59 | 2.70 | 2.30 | 2.25 | 2.80 | 1.80 | 2.53 |
| $MS$ | 1.22 | 1.81 | 1.47 | 1.05 | 0.55 | 1.05 | 1.10 | 0.80 |
| $CT$ | 3.78 | 4.99 | 4.36 | 4.00 | 5.00 | 3.30 | 3.60 | 4.15 |
| $LACAE$ | 3.51 | 3.25 | 2.62 | 2.80 | 2.80 | 3.30 | 2.30 | 3.05 |
| $VP$ | 1.31 | 0.71 | 0.06 | 0.20 | 0.20 | 0.30 | 0.70 | 0.05 |
| $RH$ | 0.65 | 0.81 | 0.43 | 0.35 | 0.35 | 0.35 | 0.30 | 0.35 |
| $albedo$ | 8.87 | 8.24 | 8.50 | 8.00 | 8.15 | 8.35 | 7.90 | 8.25 |

**Table 2: Comparison of the analysis results of the CP method and the OAT method under different scenario settings.**

In cases (1) and (2), we varied the standard deviations of the constrained factors while applying the CP method, ensuring that the data distribution remained consistent. The analysis reveals a difference in results ranging from 1.7% to 48%, with an average difference of 21%. The weighted average calculated based on sensitivity is 14%. Under varying constraints, factors exert different influences on one another due to their interactions. Consequently, the final output is shaped not only by the constraints imposed on individual factors but also by these interactions, leading to differing sensitivity analysis results. The

results demonstrate that the CP method can accurately quantify how factors contribute to the reduction of output uncertainty under varying levels of observation accuracy improvement. In contrast, the OAT method overlooks the influence of factor interactions on the outcomes, making it incapable of performing differential analyses under different observation constraints, which represents a significant limitation.

In cases (1) and (3), we applied the CP method with save standard deviations of the factors before and after the constraints.

However, the differing data distributions lead to a notable variation in analysis results, ranging from 1.3% to 95%, with an average difference of 15% and a weighted average of 4.9%. In scenarios where the volume of observational data is limited or influenced by external factors, the data cannot be guaranteed to be strictly normally distributed, and a linear trend may





appear. The presence of this linear trend, identified through the CP method, can cause substantial variability among the factors, suggesting that the uncertainty contributions of certain factors are significantly influenced by the data distribution—

an aspect not captured in the OAT method's analysis.

In cases (4), (5), and (6), we changed the selected $A$ values for the OAT method vary from -10% to +10%. The differences in the analysis results vary from 0% to 50%, with an average difference of 22% and a weighted average difference of 20%. Some factors do not act linearly on the output, which makes the sensitivity analysis results of such factors highly dependent on the choice of $A$ value. Consequently, even minor variations in the $A$ value can result in significant errors. Therefore, when

employing the OAT method for sensitivity analysis, careful attention must be given to the stability of the results.

In cases (4), (7), and (8), we changed the selected $dx$ values in the OAT method range from -50% to +100%. The differences in the analysis results vary from 0% to 300%, with an average difference of 37% and a weighted average difference of 7.8%. The sensitivity analysis outcomes for certain factors are highly sensitive to the chosen $dx$ value, and different strategies for selecting $dx$ can result in varied analysis results. Mathematically, smaller $dx$ values tend to provide a more accurate

reflection of the actual sensitivity. However, in practice, the choice of $dx$ is influenced by the model's accuracy and the distribution of actual observed data, necessitating certain trade-offs. This variability contributes to the stability issues observed in the analysis results produced by the OAT method.

A comprehensive comparison indicates that the CP method yields results that are representative of the entire data distribution. It effectively calculates differences in uncertainty contributions under various factor constraints, and distinguishes between

sensitivity results from non-normal and normal data distributions. In contrast, the OAT method provides results that are primarily relevant near a fixed value, and when data quality standards are not met, its results may exhibit poor stability.

**3.3.2 ARI Factor Uncertainty Contribution Ranking Results**

The results indicate that among the four parameter categories, the aerosol physical and chemical property parameters exhibit the most significant sensitivity. Specifically, $CRI_{shell,dry}$ demonstrates the highest sensitivity across all five sensitivity

analyses. Besides, $PNSD_{dry}$ shows strong sensitivity to both AOD and SSA, while $kappa$ exhibits pronounced sensitivity to g. When the core-shell structure model of the Mie theory is not considered, these three parameters sufficiently characterize the complex refractive index, particle number size distribution, and hygroscopic properties of aerosols. They are thus crucial for understanding the radiative forcing characteristics of aerosols.

The Mie theory parameters exhibit the second highest sensitivity. Due to the fact that changes in the aerosol mixing state can

strongly influence both AOPs and $RF_{ari}$, $CT$ demonstrate significant sensitivity across all parameters. The $LACAE$ primarily affects the aerosol's absorption characteristics, leading to strong sensitivity to SSA, which results in high sensitivity to both $RF_{ari,top}$ and $RF_{ari,bottom}$. Additionally, $MS$ influence the ratio of LAC to coating materials, significantly affecting the shape of the aerosol particle number size distribution; consequently, it show considerable sensitivity to g.



The sensitivity of LAC parameters ranks third. Specifically, the parameters $n_{BC}$, $k_{BC}$, and $\rho_{BC}$ influence the aerosol's CRI,

while $LACPNSD$ affects both the LAC particle size spectrum distribution and the aerosol mixing state. The results indicate that, as the primary absorptive component of aerosols, the complex refractive index of LAC significantly impacts aerosol absorption characteristics, leading to a pronounced effect on SSA. This, in turn, results in strong sensitivity of both $RF_{ari,top}$ and $RF_{ari,bottom}$. Furthermore, $LACPNSD$ exhibits the highest sensitivity among the LAC parameters, highlighting the substantial influence of the aerosol mixing state on $RF_{ari}$. Notably, the limited availability of observations for LAC

parameters may contribute to considerable evaluation errors.

This study also examines the influence of environmental factors on $RF_{ari,top}$ and $RF_{ari,bottom}$. Three environmental parameters are discussed: $VD$, $RH$, and $albedo$. Among these, both $RH$ and $albedo$ demonstrate significant sensitivity to $RF_{ari,top}$ and $RF_{ari,bottom}$. These environmental parameters effectively characterize the impact of boundary layer characteristics, vertical humidity profiles, and surface conditions on $RF_{ari}$. The results indicate that the effects of

environmental factors on $RF_{ari}$ are comparable to those of the aerosol's own radiative characteristics and should not be overlooked in research calculations. Notably, the sensitivity of $VD$ to radiative forcing differs markedly between the top of the atmosphere and the surface. While $VD$ has minimal impact on $RF_{ari,top}$, it exhibits considerable sensitivity to $RF_{ari,bottom}$, suggesting that the type of boundary layer significantly influences surface heating rates but has a lesser effect on the overall ground-atmosphere radiation budget.

**4 Summary**

The evaluation results of $RF_{ari}$ indicate significant uncertainty, with notable discrepancies between observational and simulation outcomes. This discrepancy contributes to the overall uncertainty in climate sensitivity assessments. This study introduces a novel method for analyzing factor uncertainty contributions, enabling a quantitative ranking of the contributions from various factors and addressing the extent to which enhancing observational accuracy can reduce result uncertainty.

Additionally, through a comprehensive analysis of the factors of ARI system, this research identifies several previously overlooked factors of importance, providing valuable insights for aerosol observation projects and model settings.

This study analyzes the advantages and disadvantages of the OAT method, which is currently the most commonly used method in the ARI factor uncertainty contribution analysis. While the OAT method facilitates rapid quantification of factor sensitivity, it also brings the defects of high data quality requirements and poor stability. For ARI systems, the OAT method

faces challenges due to the low accuracy of observational data, potential linear trends in the data distribution, and significant interactions among factors. Additionally, this method only provides sensitivity results close to fixed values, making it less reliable both mathematically and physically. As a result, the OAT method may produce substantial errors. To address these challenges, this work introduces a new analysis method, CP, designed for sensitivity analysis of both input and output uncertainties. This method is universally applicable to all multi-factor systems. Unlike traditional methods that focus on

sensitivity near specific values, CP is based on the collective data distribution, providing a more comprehensive representation. It directly addresses how improvements in data accuracy can enhance result certainty, offering practical insights. Additionally, CP can assess differences in factor uncertainty contributions under various observational constraints and with non-normally distributed data, broadening its applicability.

This study employs Mie theory to calculate the AOPs and utilizes the SBDART radiative transfer model to simulate $RF_{ari}$.

The factors across all physical processes are categorized based on their modes of influence into four groups: aerosol physical and chemical property parameters, Mie theory parameters, LAC parameters, and environmental parameters. Each category undergoes a separate uncertainty contribution analysis. The results reveal that among all factors, $CRI_{shell,dry}$ holds the highest importance. Additionally, both the LAC parameters and Mie theoretical parameters demonstrate high uncertainty contributions, indicating that the scattering and absorptive properties of aerosols, along with the assumptions inherent in Mie

theory, substantially influence $RF_{ari}$. Due to the challenges associated with direct observation, these factors have been insufficiently addressed in routine observation projects and model settings. To mitigate the high uncertainty associated with $RF_{ari}$ evaluations, it is imperative to focus attention on these critical factors.

**Code availability**

The code used in this work can be found at https://github.com/sGhotHe/Code.

**Data Availability**

The data used in this work can be found at https://github.com/sGhotHe/Data.

**Author Contribution**

Bishuo He: Data curation, formal analysis, investigation, methodology, resources, software, validation, visualization, writing – original draft preparation.

Chunsheng Zhao: Conceptualization, funding acquisition, project administration, supervision, writing – review & editing.

**Competing Interests**

The authors declare that they have no conflict of interest.



**Acknowledgements**

We thank Gang Zhao for providing aerosol and black carbon observation data for Mie model calculations. We thank Weilun
Zhao and Jie Qiu for their help in analyzing some calculation results. This study was supported by the National Natural
Science Foundation of China (Grant No.42275070).

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
