# Peer review of "A Novel Method to Quantify the Uncertainty Contribution of Aerosol-Radiative Interaction Factors"

_EGUsphere, 2024_

## Author Comment (AC2)

Thank you for your constructive and insightful comment. We are grateful for your recognition of the CP method proposed in this work. We have carefully addressed all comments and suggestions. Below are our responses and modifications to the reviewer's concerns.

**Response:**

In this review, the reviewer expressed concerns about the data and results, suggesting a comparison with findings from other literature. This work aims to provide a more detailed assessment of the contribution of factor uncertainty to aerosol-radiation interactions through an in-depth factor analysis. Therefore, we utilize a large dataset of near-ground aerosol property observations, including the complex refractive index, hygroscopicity, density, and aerosol particle size distribution. All data are derived from field observations conducted by our research group in North China using self-built observational instrument systems, as described in Table 1 of the manuscript.

Field experiments to gather such extensive aerosol property data in the same region require considerable time and instrument resources. In contrast, previous studies or standard observation sites rarely provide sufficient data to match the scope of our work, which we identified during our literature and data research. As a result, we did not incorporate additional verified data into the experiments. While this limits the robustness of our results to some extent, as discussed in Section 3.1, our objective is to apply the CP method within the factor analysis framework we have developed and compare it with the OAT method to verify the characteristics of the CP method. As the reviewer noted, the results obtained in this study are not universally applicable but rather specific to the environmental settings in this work.

The results produced by the CP method depend critically on the performance of the model used. In other words, they are an evaluation of the model. The closer the model's description of real physical processes, the more accurate the results obtained by the CP method will be. With this in mind, we employed the Mie theory model to accurately describe the single-particle radiation process and the SBDART radiation transfer model to focus on the radiation transfer process, aiming to minimize model-induced uncertainty. The use of the Mie theory model also allows for some of the conclusions in this work to be directly verified, such as the identification of the aerosol complex refractive index and coating thickness as highly sensitive factors.

One important consideration is that the conclusions obtained by the CP method are also influenced by the distribution of the input aerosol property data. Using the same model framework with aerosol property data from typical urban pollution aerosols in regions outside North China for sensitivity analysis may yield significantly different results, as discussed in Section 2.3. In summary, obtaining universal conclusions requires more accurate models and representative, detailed aerosol property data. We plan to address these aspects in our future work. Additionally, we hope to use our CP method for a detailed factor uncertainty contribution analysis of the black carbon absorption enhancement phenomenon to resolve the current discrepancies between theory and observation regarding this phenomenon.

We will organize the above discussion and incorporate it into the **Conclusions and Discussions** section of the manuscript.

**Modification**:

In the Conclusions and Discussions section, add new paragraphs to answer concerns about the

robustness of the conclusions reached in this work (Page 15, lines 398-406 in revised manuscript): "*The quantitative ranking results of factor uncertainty obtained in this study using the CP method are highly dependent on the model employed and the distribution of input aerosol optical property data. The more accurately the model describes the actual physical processes and the higher the precision of the aerosol optical property data, the closer the evaluation results from the CP method will align with observed values. Therefore, this study utilizes the Mie theoretical model and the SBDART radiation mechanism model, which provide a more accurate representation of the single-particle radiation process, along with field experimental data from direct observations of aerosol optical properties. This approach aims to minimize the uncertainty introduced by the model and data accuracy. Using different model frameworks or aerosol optical property data from polluted aerosols in cities outside of North China to conduct the same factor sensitivity analysis may yield significantly different results.*"